

# Investigating diversity of pathogenic microbes in commercial bait trade water

Andrew R. Mahon[1], Dean J. Horton[1], Deric R. Learman[1], Lucas R. Nathan[2] and Christopher L. Jerde[3]

[1] Department of Biology, Institute for Great Lakes Research, Central Michigan University, Mount Pleasant, MI, United States of America
[2] Department of Natural Resources and the Environment, University of Connecticut, Storrs, CT, United States of America
[3] Marine Science Institute, University of California, Santa Barbara, Santa Barbara, CA, United States of America

## ABSTRACT

The recreational bait trade is a potential pathway for pathogen introduction and spread when anglers dump bait shop sourced water into aquatic systems. Despite this possibility, and previous recognition of the importance of the bait trade in the spread of aquatic invasive species (AIS), to date there has been no region wide survey documenting pathogens in retail bait shops. In this study, we analyzed 96 environmental DNA samples from retail bait shops around the Great Lakes region to identify pathogens, targeting the V4 hypervariable region of the 16S rRNA gene. Additionally, we used samples from one site in Lake Michigan as a comparison to pathogen diversity and abundance in natural aquatic systems. Our results identified nine different groups of pathogens in the bait shop samples, including those that pose risks to both humans and fish species. Compared to wild sourced samples, the bait shops had higher relative abundance and greater taxonomic diversity. These findings suggest that the bait trade represents a potentially important pathway that could introduce and spread pathogens throughout the Great Lakes region. Improving pathogen screening and angler outreach should be used in combination to aid in preventing the future spread of high risk pathogens.

## INTRODUCTION

With over 30 million anglers in the United States and Canada, and with many of them using live bait in the form small fish (*USDI, 2011*; *DFO , 2012*), there is a significant risk of invasive species introduction and spread through the commercial bait trade vector (*Drake & Mandrak, 2014*). This is particularly alarming when commercial bait retailers are contaminated with invasive fish, such as Goldfish (*Carassius auratus*), Round Goby (*Neogobius melanostomus*), Eurasian Rudd (*Scardinius erythrophthalmus*), and Silver Carp (*Hypophthalmichthys molitrix*) (*Nathan et al., 2015*). Invasion risk significantly increases when anglers dump unused bait and water into the lakes and rivers at the end of a fishing day (*Drake & Mandrak, 2014*). However, this angling behavior has the potential to introduce more than just invasive fish species into new areas. *Smith et al. (2012)* revealed

Corresponding author
Andrew R. Mahon,
mahon2a@cmich.edu

the water in which ornamental fish are transported contains a unique biota of fish and human pathogens beyond those found in the fish themselves. Additionally, the diversity of pathogens within wild and cultured baitfish is well studied (*Goodwin et al., 2004*; *Lowry & Smith, 2007*). As such, it is reasonable to suspect that water which contains bait fish could also serve as a reservoir for pathogenic bacterial species. This raises the question, what pathogens are found in the bait bucket water?

Pathogens have the potential to be very damaging to human and wildlife health (*Daszak, Cunningham & Hyatt, 2000*) and economically costly (*Jenkins, 2012*). With respect to fisheries, pathogens can be spread between commercial operations and wild populations, resulting in costly damages. Such was the case with amplified sea lice densities from farmed Atlantic Salmon leading to the decline of native Coho and Pink Salmon populations in British Columbia, Canada (*Krkošek et al., 2007*; *Krkošek et al., 2009*; *Krkošek et al., 2011*). The damages become more acute when the pathogens in question are generalists and spread throughout a valuable fishery, such as with viral hemorrhagic septicemia virus (VHSV) spread throughout the Laurentian Great Lakes (*Rothlisberger et al., 2010*; *Escobar et al., 2017*). While VHSV is a known pathogen already in the Great Lakes region, the identities and impact of other pathogens are largely unknown.

In the summers of 2012 and 2013, we visited over 500 bait shops across the U.S. Great Lakes states of Minnesota, Wisconsin, Illinois, Indiana, Michigan, Ohio, Pennsylvania and New York to collect water samples from commercial bait tanks for use in environmental DNA (eDNA) screening of invasive species (*Mahon, Nathan & Jerde, 2014*; *Nathan et al., 2015*). Our hypothesis was that if invasive species are in a bait tank, then the water would contain DNA from sloughed tissue, cells, and organelles in the water, which could be filtered, extracted, and screened using molecular tools to detect the invasive species (*Ficetola et al., 2008*; *Jerde et al., 2011*; *Simmons et al., 2016*). Additional to our initial hypotheses, these extracted eDNA samples also contain DNA from all organisms in the water, including potential pathogens, similar to those evaluated by *Smith et al. (2012)* who screened water samples for pathogens in the water of imported, ornamental fish.

Here, we repurpose the eDNA samples previously collected from commercial bait shops in the search for Great Lakes invasive fish species and analyze them using similar methods to those employed by *Smith et al. (2012)* to document putative pathogens. Our goal was to identify pathogenic species present in the samples, compare the diversity and abudance of bait shop sourced pathogens to Great Lake sourced pathogens, and evaluate the potential threat of unique, bait sourced pathogens being spread in the Laurentian Great Lakes in a manner similar to that documented for invasive species.

## METHODS

### Sample collection and DNA extraction

Two-liter water samples were collected from the bait holding tanks in commercial bait shops from each of the states in the Laurentian Great Lakes basin (Table 1; Fig. 1; for additional collection details see *Nathan et al. (2015)*). Samples were filtered through ∼1.5 µm glass fiber filter paper within 24 h of their collection. DNA was extracted from filtered samples

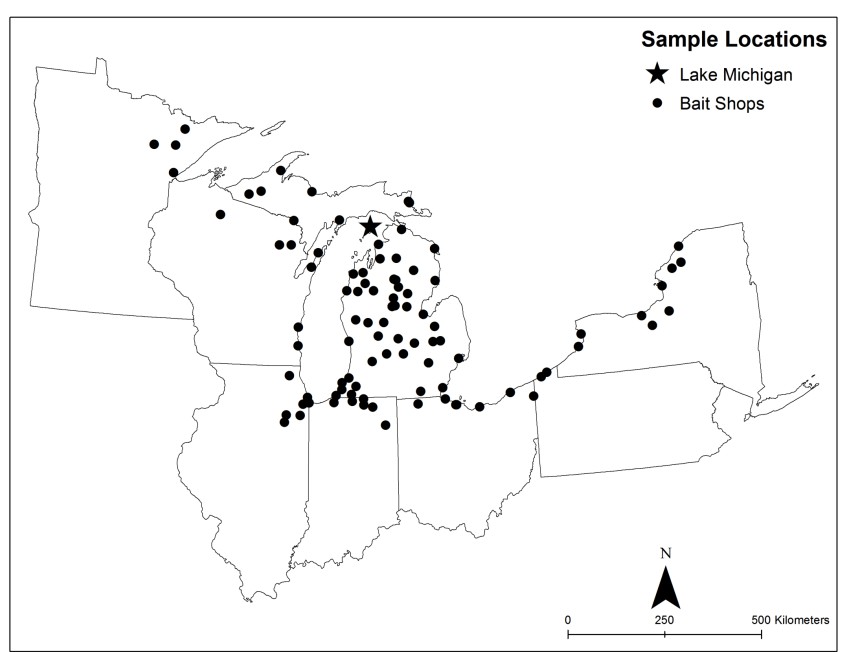

**Figure 1** **Map of collection locations for samples used in this study.** Collection locations of bait water samples collected from commercial vendors utilized in this study. Included are shop locations (black circles) and the sampling location for the Lake Michigan water sample (black star) included in the dataset.

**Table 1** **Sample collection locations.** Collection location (by US state) and number of bait shop DNA samples used in the study.

| State | Number of samples |
| --- | --- |
| IL | 6 |
| IN | 7 |
| MI | 53 |
| MN | 4 |
| NY | 9 |
| OH | 7 |
| PA | 2 |
| WI | 8 |

using a MOBio PowerWater DNA Isolation Kit (MoBio Laboratories, Carlsbad, CA, USA) following manufacturer recommendations. All samples for this study were collected and analyzed using previously described quality assurance protocols (*Mahon et al., 2010*; *Jerde et al., 2011*; *Mahon et al., 2013*; *Jerde et al., 2013*; *Nathan et al., 2015*). Samples were chosen for analyses based on two factors: (a) available DNA remaining from previous studies (*Mahon, Nathan & Jerde, 2014*; *Nathan et al., 2015*) and (b) proportional number of samples (out of 96 total) based on availability from each Great Lakes state where samples were collected. Upon consideration of those factors, samples were then randomly chosen for inclusion in this study. Total number of samples used in this study are listed in Table 1
by collection location. Because our previous collection site numbers (i.e., bait shops) were not equal from all Great Lakes states, proportionally, some states had more samples included in this investigation than others (Table 1). We chose to restrict our location information for individual bait shops (names, street addresses) where sample collections were taken to the US state to conceal the identity of individual vendors.

Along with investigating water samples from commercial bait vendors, we included sequence data collected from a location in northern Lake Michigan to provide a point of comparison between water samples collected in commercial bait shops and in natural Great Lakes ecosystems (*Hengy et al., 2017*; Fig. 1). Five samples from one collection site were included in this study to serve as a comparison to our bait shop sequencing data. Although the Lake Michigan samples likely do not represent the true pathogenic diversity in the entire Great Lakes region, we included the wild samples to provide a comparison to the potential differences between wild and bait sourced pathogens.

### Genetic sequencing

Genomic DNA extracted from each of the 96 eDNA samples was sent to the Michigan State University Research Technology Support Facility for microbial amplicon sequencing. Amplicon sequencing libraries targeting the V4 hypervariable region of the 16S rRNA gene (515f/806r) were made following the protocol described by *Kozich et al. (2013)*. After PCR amplification, resulting amplicon products were normalized using Invitrogen's SequalPrep DNA normalization plates, pooled and purified. Pooled amplicons were validated and quantified using Qubit dsDNA, Caliper LabChipGX DNA, and Kapa Biosystems Illumina Library Quantification qPCR assays. The pool of samples was then loaded on an Illumina MiSeq flow cell (v2) and sequenced in a $2 \times 250$ bp paired end format with a 500 cycle v2 reagent cartridge. Base calling was done by Illumina Real Time Analysis (RTA) v1.18.54 and the sequencing output was demultiplexed and converted to FastQ format using Bcl2fastq v1.8.4.

### Data filtering, QAQC, and analyses

Sequences were screened for quality using MOTHUR version 1.35.1 (*Schloss et al., 2009*) following the MiSeq SOP (https://www.mothur.org). Paired-end reads were assembled into contigs and were retained if they were between 251 and 254 bp in length, contained $\leq 8$ homopolymers, and lacked ambiguous bases. Sequences were then aligned against the Silva (v. 119) rRNA database (*Quast et al., 2013*) and chimeric DNA sequences were screened for and removed with UCHIME (*Edgar et al., 2011*). Sequences were classified using the Ribosomal Database Project (RDP) (training set v9; *Cole et al., 2014*). Reads identified as chloroplast, mitochondrial, or eukaryotic DNA, as well as those with unknown classifications, were removed from the dataset. Operational Taxonomic Units (OTUs) were clustered using a threshold of 0.03 sequence dissimilarity. Additionally, any OTUs that were represented less than twice in the dataset were removed as a conservative measure. Following data processing, we then screened our results for a targeted group of potential pathogens similar to those noted by *Smith et al. (2012)* (Table 2) using standard NCBI BLAST searches (*Altschul et al., 1990*). The search for putative pathogens was limited to

**Table 2  Pathogens screened.** List of bacterial pathogen groups that were searched for in our resulting data. Note that while not all members of each group listed are known to cause the effects listed, these are "worst case" scenarios for that genus/group.

| Pathogen | Some harmful effects caused by members of group |
| --- | --- |
| *Vibrio* | Some species can cause gastroenteritis, septicemia, cholera |
| *Legionella* | Legionnaires disease, Pontiac fever |
| *Mycobacterium* | Tuberculosis, leprosy |
| *Coxiella* | Q fever |
| *Campylobacter* | Campylobacteriosis (gastrointestinal infection) |
| *Francisella* | Tularemia (rabbit fever), septicemia and invasive systemic infections |
| *Plesiomonas* | Gasteroenteritis |
| *Flavobacterium* | Bacterial cold water disease on salmonids, rainbow trout fry disease on rainbow trouts, cotton-wool disease on freshwater fishes, the bacterial gill disease on trouts. |
| *Salmonella* | Typhoid fever, paratyphoid fever, and food poisoning |
| *Giardia* | Giardiasis |
| *Shigella* | Shigellosis, dissentary |
| *Aeromonas* | Gastroenteritis and wound infections, with or without bacteremia. |

those that were used in the study, *Smith et al. (2012)*. This is not an exhaustive list of pathogens of concern in the Great Lakes, however, it allows for a comparison of putative pathogens that are related to the unique ecosystem found in bait tank water. The search did include some known fecal indicator bacteria, *E. coli*, *Enterococcus*, *Staphylococcus*, and *Bacteroides* (see reviews, *Sinigalliano et al., 2010*; *McLellan, Fisher & Newton, 2015*). The data described here did not contain any OTUs that had enough genetic resolution to be classed as *E. coli*.

The same approach for processing and analyzing the dataset, from sample collection through data filtering, was applied to samples collected from Lake Michigan (*Hengy et al., 2017*). Briefly, five samples were collected and filtered (0.2 μm) from St. John's Bay (Lake Michigan), Beaver Island, Michigan, USA. DNA was extracted and sequenced targeting the V4 region of the 16s rRNA gene as described above for the commercial bait shop samples. This set of samples was chosen for a comparision site as it used the same amplicon region and sequcing plateform as used in this study and was also available immediately for our use in this study. Comparing different amplicon regions and sequencing plateforms have been shown to be difficlult and could provide misleading information and conclusions (e.g., *Claesson et al., 2010*). Additionally, we used a Chi-square test to evaluate independence between bait shop sourced and Lake Michigan sourced pathogen sequence counts. All tests were performed in Mathematica (*Wolfram Research Inc, 2017*).

## RESULTS

Sequence data associated with this study are available on the MG-RAST database (*Meyer et al., 2008*) under accession numbers mgm4791794.3 –mgm4791986.3 and as referenced in *Hengy et al. (2017)*. From our resulting sequence data of the V4 hypervariable region of the 16S rRNA gene (515f/806r), a total of 1,594,572 sequence reads (of 8,082,960 total assembled sequences) matched the nine targeted groups listed in Table 2. Of these,

**Table 3  Sequence reads bait shops.** Reads and OTUs of targeted pathogen groups for bait sourced samples and their virulence/pathogenic effects regardless of their origin. Note that Top/Notable matches from BLAST results were for all top matches >98% similar for the OTUs found.

| | Total reads in all samples | Highest # of reads in single sample (location) | Total OTUs found in dataset from all shops | Present in # of samples (and % of samples) | Top/Notable matches (disease) |
|---|---|---|---|---|---|
| *Vibrio* | 20,977 | 6,406 (MI) | 1 | 65 (67.71%) | *V. cholerae* (cholera) , *V. anguillarum* (cultured salmon pathogen) |
| *Legionella* | 26,419 | 7,264 (WI) | 141 | 86 (89.58%) | *L. maceachernii* (pneumonia), *L. pneumophilia* (Legionnaires disease), *L. micdadei* (Pontiac fever), *L. longbeachae* (Pontiac fever) |
| *Mycobacterium* | 164,190 | 51,052 (WI) | 37 | 93 (96.88%) | *M. tuberculosis* (tuberculosis), *M. bovis* (TB in cattle), *M. lepromatosis* (leprosy), *M. microti* (other mammal TB), *M. tusciae* (chronic fibrosis from tap water), *M. mucogenicum* (BSL 2, skin infections) |
| *Coxiella* | 95 | 37 (IN) | 6 | 7 (7.29%) | Q fever (closest at 95% match) |
| *Campylobacter* | 1,414 | 503 (WI) | 2 | 5 (5.21%) | *C. consisus* (intestinal disease), *C. gracilis* (intestinal disease) |
| *Francisella* | 1,164 | 485 (MI) | 1 | 17 (17.71%) | *F. philomiragia* (rare human infection) |
| *Plesiomonas* | 17 | 17 (MI) | 1 | 1 (1.04%) | Unknown strain. |
| *Flavobacterium* | 1,173,491 | 64,265 (MN) | 400 | 100 (100%) | *F. psychrophilum* (bacterial cold water disease in salmonids, rainbow trout fry disease), *F. columnare* (cotton-wool disease in freshwater fish), *F. branchiophilum* (bacterial gill disease in trout) |
| *Salmonella* | – | – | – | – | Not present |
| *Giardia* | – | – | – | – | Not present |
| *Shigella* | – | – | – | – | Not present |
| *Aeromonas* | 206,805 | 10,426 (MI) | 24 | 95 (98.95%) | *A. veronii* (human pathogen), *A. salmonicida* (virulent salmon pathogen), *A. schubertii* (infection of Chana argus!) |

*Flavobacterium* was the most diverse (400 OTUs) and most abundant (1,173,491 sequences) taxonomic group of putative pathogens (Table 3). Additionally, *Flavobacterium* was present in all of the eDNA samples processed and sequenced as a part of this effort. Least common in our resulting data was *Plesiomonas*, with a total of 17 reads found in the water sample from a single commercial bait shop in Michigan and the resulting BLAST search matches were to an unknown strain. Further, our results found that both potential human and fish pathogens were present in water samples collected in the commercial bait trade (Table 3). While we restricted our study to pathogens in trade (see *Smith et al., 2012*), our data from bait shop samples did show the presence of three putative fecal indicator bacteria: *Bacteroides* (13 OTUs with 9828 total reads), *Staphylococcus* (one OTU with 56356 total reads), and *Enterococcus* (one OTU with 3706 total reads). Future investigations on these and other comparable datasets should investigate these organisms further.

Compared to bait samples, the five water samples collected from one location in northern Lake Michigan had only five of nine of our chosen target groups present when sequenced and analyzed in same fashion (Table 4). Additionally, numbers of OTUs for Lake Michigan targeted groups were significantly lower (ranging from 1-39 total OTUs; Table 4). The Chi-square test indicated the distribution of pathogens was different between Lake Michigan and Bait shops ($p$-value <0.001, $d.f. = 8$).

**Table 4  Sequence data Lake MI.** Reads and OTUs of targeted pathogen groups for the five Lake Michigan samples (one collection location; *Hengy et al., 2017*) and their virulence/pathogenic effects regardless of their origin. Note that Top/Notable matches from BLAST results were for all top matches >98% similar for the OTUs found.

| | Total # of reads in all samples | Highest # of reads in single sample/ location | Total OTUs found in dataset from all samples | Present in # of samples (and % of samples) | Top/Notable matches (disease) |
|---|---|---|---|---|---|
| *Vibrio* | 0 | 0 | 0 | 0 | Not present |
| *Legionella* | 16 | 4 | 11 | 5 (100%) | *L. longbeachae, L. wadsworthii, L. santicrucis* |
| *Mycobacterium* | 9 | 4 | 6 | 3 (60%) | n/a |
| *Coxiella* | 1 | 1 | 1 | 1 (20%) | |
| *Campylobacter* | 0 | 0 | 0 | 0 | Not present |
| *Francisella* | 0 | 0 | 0 | 0 | Not present |
| *Plesiomonas* | 0 | 0 | 0 | 0 | Not present |
| *Flavobacterium* | 1,379 | 915 | 39 | 5 (100%) | *Pseudomonas veronii, Pseudomonas gessardi, Pseudomonas sp., Pseudomonas fluorescens* |
| *Salmonella* | – | – | – | – | Not present |
| *Giardia* | – | – | – | – | Not present |
| *Shigella* | – | – | – | – | Not present |
| *Aeromonas* | 46 | 33 | 1 | 5 (100%) | *Aeromonas jandaei, Aeromonas allosaccharophilia, Aeromonas bivalvium, Aeromonas molluscorum, Aeromonas caviae, Aeromonas salmonicida*, others (all 100%) A.c. causes necroticizing fasciatus |

## DISCUSSION

In this study, we documented the presence of human and fish pathogens in commercial bait retailers in the Great Lakes region using genomic surveillance. Compared to a sample sourced from Lake Michigan, bait samples had higher counts and higher diversity of multiple groups of pathogens. Bait samples collected for this study even found the presence of human fecal indicator bacteria (*Bacteroides, Staphylococcus*, and *Enterococcus*). Given the number of recreational anglers that use live bait and potentially dispose of bait bucket water into aqutic systems, the bait trade represents a potential vector for introducing and spreading pathogens throughout the Great Lakes. Along with this, angler's use of bait presents the possibility of contact with the bait tank water increasing risk of exposure to these potential pathogens. While the virulence of these organisms remains unknown, this still represents a distinct possibility of transfer, spread, and/or infection when they are present in the system.

While most any water sample, be it from the holding water of an ornamental fish (*Smith et al., 2012*) or a commercial bait shop, local pond, drinking water source, or from a Great Lake, is expected to have some pathogenic biota, clearly there is a difference in the distribution and diversity of pathogens contained within samples. Our dataset found no evidence of *Vibrio, Campylobater, Francisella*, or *Pleasiomonas* bacterial strains in the limited number of samples from Lake Michigan water we sampled and sequenced, yet they comprised 1.5% of the sequenced pathogens in bait shop sourced water. While admittedly

rare in our bait shop samples, these pathogens may be thriving or at some ecological advantage in bait shops or they may be at undetectable levels in the natural system. In contrast, OTUs similar to *Aeromonas* pathogens were nearly four times more abundant (as a percentage) in bait shops than in Lake Michigan sourced water. This again demonstrates an important point about these data, where a high percentage, i.e., dominant, microbes in bait water are potential pathogens, and in our Lake Michigan sampling, dominant microbes were likely non-pathogens. A study by *Fujimoto et al. (2016)* on lake water did not find OTUs classified as *Vibrio*, *Mycobacterium*, *Aeromonas*, or *Pleasiomonas* but did find one OTU that can be classeifed as *Campylobater* and *Francisella*. However, these were very rare as the study defined a total of 158,900 OTUs (*Fujimoto et al., 2016*). Although, in this study, the wild sourced Lake Michigan collections (from five samples sequenced and included here), came from a relatively limited spatial extent, which likely represent a fraction of the true pathogenic diversity in the Great Lakes, the contrast between the wild and bait shop samples suggests a substantial variation between the two sources. Future investigations should include a wider sampling effort from throughout the basin to make a more direct comparison between these two sources of water samples.

## CONCLUSIONS

Detection and identification of pathogens is not new to science with over 19,000 articles published on pathogen detection in the last 10 years (from 2008-2018; IS Web of Science, query on 3/15/18). Researchers have previously reported on pathogen identification from areas outside of water and the environment, including the food industry, defense, and clinical applications (*Lazcka, Campo & Muñoz, 2007*). Analytical methods for these industries include traditional screening procedures (culturing and colony counts) through molecular methods and biosensors (*Lazcka, Campo & Muñoz, 2007*). However, the role bait water plays in pathogen transmission is unclear.

In the United States, there are approximately 33 million people that participate in recreational fishing (16 years of age and older) that account for an industry of over $48 billion annually and approximately 828,000 jobs (*Southwick, 2012*). Within this, the access to and the use of live bait from commercial shops presents a need to ensure the safety of participants. Spread of these pathogens is an ongoing important problem in the bait industry. Despite the concern over pathogens in the industry, actions towards prevention of pathogen spread by commercial vendors do not always address the issue (*Connelly et al., 2018*).

In this study, we note a number of potentially harmful pathogens in samples fully accessible to the public. However, there are a number of caveats to this. First, potential pathogens, albeit present in the samples as noted, could be at low levels and may never become virulent. Additionally, the pathogenicity of the documented OTUs was not specifically tested in this study. Second, there is no guarantee of spread from source (e.g., commercial shop) to additional sites, or that bait water bacteria could become harmful to humans as the pathogens may become damaged during transport or die when released into the environment, a similar observation to that of the importance of propagule pressure in

the invasive species literature (*Lockwood, Cassey & Blackburn, 2005*). However, while there are no guarantees for spread and/or infection, the likely repeated introduction of these potentially dangerous strains of organisms to the environment and as documented in the invasive species literature, even low probability survival and arrival can ultimately lead to establishment and damages of invasive species (*Jerde & Lewis, 2007*; *Jerde, Bampfylde & Lewis, 2009*). Future screening and monitoring is needed to ensure safety for millions of participants in recreational fishing annually and also to the ecosystems where these harmful pathogens could be spread.

## ACKNOWLEDGEMENTS

This paper is Contribution Number 104 of the Central Michigan University Institute for Great Lakes Research.

### Funding

This project was supported by a Great Lakes Restoration Initiative (GLRI) grant (EPA-R5-GL2011–1) to Andrew R. Mahon and Christopher L. Jerde. The funders had no role in study design, data collection and analysis, decision to publish, or preparation of the manuscript.

### Grant Disclosures

The following grant information was disclosed by the authors:
Great Lakes Restoration Initiative (GLRI): EPA-R5-GL2011–1.

### Competing Interests

The authors declare there are no competing interests.

### Author Contributions

- Andrew R. Mahon conceived and designed the experiments, performed the experiments, analyzed the data, contributed reagents/materials/analysis tools, prepared figures and/or tables, authored or reviewed drafts of the paper, approved the final draft.
- Dean J. Horton analyzed the data, prepared figures and/or tables, authored or reviewed drafts of the paper, approved the final draft.
- Deric R. Learman and Christopher L. Jerde conceived and designed the experiments, analyzed the data, contributed reagents/materials/analysis tools, prepared figures and/or tables, authored or reviewed drafts of the paper, approved the final draft.
- Lucas R. Nathan performed the experiments, analyzed the data, prepared figures and/or tables, authored or reviewed drafts of the paper, approved the final draft.

## Data Availability

Sequence data associated with this study are available on the MG-RAST database (*Meyer et al., 2008*) under accession numbers mgm4791794.3–mgm4791986.3 and as referenced in *Hengy et al. (2017)*.

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
