# Peer review of "Investigating diversity of pathogenic microbes in commercial bait trade water"

_PeerJ, doi:10.7717/peerj.5468_

## Round 0.1 · original submission · Minor Revisions

Dear Authors,

The reviewers have commented on your above paper. They indicated that it is not acceptable for publication in its present form.

However, if you feel that you can suitably address the reviewers' comments, with special consideration to the comments about the experimental design, I invite you to revise and resubmit your manuscript.

Best regards

Reviewer 1 ·

Basic reporting

This article describes findings from a microbial DNA analysis of 96 water samples collected from holding tanks in bait shops located in eight states in the Great Lakes Region. Findings were compared to data from a single location near Beaver Island which is located in the northernmost area of Lake Michigan. This article is well-written and the abstract appropriately summarizes the research hypothesis, methods and findings. The number and selection of articles cited seems appropriate.

Experimental design

The research described is original and the question being addressed is relevant to the quality and protection of the Great Lakes ecosystem. The experimental design used to select water samples for analysis appears valid, however it is unclear why so many of the bait shops were located in Michigan and why so many of the Michigan locations appear to be more than 50 miles from the nearest Great Lake shoreline. The sampling of the ‘control’ area in northern Lake Michigan is not described, however an earlier article by Hengy et al. is referenced.

This section includes a detailed description of the genetic analysis. This is not an area I am qualified to evaluate and should be reviewed by others who work in this area.

The genetic analysis was limited to a relatively small number of pathogens. A USGS report on pathogens found at beaches on the Great Lakes assessed common fecal bacteria including E. coli, Enterococci, Cryptosporidium, and norovirus, among others of public health concern (https://pubs.usgs.gov/fs/2013/3071/pdf/fs2013-3071.pdf). Yet none of these were included in this article which summarizes findings for 12 pathogens, many of which seem less likely to be of concern to the Great Lakes. The selection of pathogens merits additional explanation.

Validity of the findings

The author’s findings are interesting and appear valid.

Additional comments

This is an interesting article that describes original research. However, there are limitations in the study design the most important of which seems to be the selection and number of pathogens included in the analysis. The relevance of the authors’ findings to the condition of the Great Lakes ecosystem, the safety of public source waters, and to the health of wildlife, domestic animals, anglers and beachgoers is not clear. For example, genetic sequencing does not indicate whether the identified organisms are viable or whether a pathogen can survive and replicate in the cold waters of these lakes. Perhaps the discussion section could be expanded to include ideas for future research that would answer questions regarding the potential for the identified pathogens, as well as pathogens not included in this study but likely to be found in bait shop tanks, to survive in the Great Lakes and infect aquatic biota, public water supplies, animals, swimmers and consumers of fish from these lakes.

Minor editorial suggestions:
1. In line 103, the word ‘identify’ should be changed to ‘identity’.
2. The numbers in Table 1 do not exactly match the dots of the map in Fig 1. Perhaps the dots in Fig 1 are too large to show all sampling sites.
3. Tables 2, 3 and 4 should be revised to be more easily compared. I suggest listing pathogens in alphabetical order and use of the same font for each table. Tables 3 and 4 should show the number of positive samples. Percent can be shown in parentheses, but this likely isn’t necessary due to the small number of samples included in this study.

·

Basic reporting

No comments

Experimental design

Really, my only concern with the manuscript is that the authors use a single wild location in Lake Michigan as a basis for comparison of all the bait shops spread around the region. I don't think it's appropriate to make broad statements about the amount of pathogen signal present in bait water vs lake water when using only one location in the wild. I suggest using at least a few more lake sites, possibly from a few other of the Great Lakes.

Validity of the findings

As stated above, the findings need more grounding against natural/field conditions. Otherwise the science of the manuscript is sound.

---

## Round 0.2 · accepted · Accept

Thank you for improving your manuscript according to the reviewers' suggestions.

Thank you for submitting your work to this journal.

#